# Mouse rods signal through gap junctions with cones

**Sabrina Asteriti[1], Claudia Gargini[2], Lorenzo Cangiano[1]\***

[1]Department of Translational Research, University of Pisa, Pisa, Italy; [2]Department of Pharmacy, University of Pisa, Pisa, Italy

**Abstract** Rod and cone photoreceptors are coupled by gap junctions (GJs), relatively large channels able to mediate both electrical and molecular communication. Despite their critical location in our visual system and evidence that they are dynamically gated for dark/light adaptation, the full impact that rod–cone GJs can have on cone function is not known. We recorded the photovoltage of mouse cones and found that the initial level of rod input increased spontaneously after obtaining intracellular access. This process allowed us to explore the underlying coupling capacity to rods, revealing that fully coupled cones acquire a striking rod-like phenotype. Calcium, a candidate mediator of the coupling process, does not appear to be involved on the cone side of the junctional channels. Our findings show that the anatomical substrate is adequate for rod–cone coupling to play an important role in vision and, possibly, in biochemical signaling among photoreceptors.

## Introduction

In darkness, sparse single photon signals are relayed to retinal ganglion cells via a high gain/high convergence pathway formed by rods, rod bipolar cells, amacrine AII cells, and cone bipolar cells (*Nelson, 1982*; *Dunn et al., 2006*). However, rod signals can bypass rod bipolar cells and enter directly into the cone pathway (*DeVries and Baylor, 1995*). The earliest opportunity for this crossover is represented by rod–cone gap junctions (GJs) (*Raviola and Gilula, 1973*; *Tsukamoto et al., 2001*). GJs are assemblies of channels made by the docking of pairs of connexon hemichannels on adjacent cells, each formed by six connexin subunits. Cones contact nearby photoreceptors mainly at the tips of thin *telodendria*, which emerge from their synaptic pedicles, where they express connexin isoform 36 (Cx36) (*Lee et al., 2003*; *O'Brien et al., 2012*), while the isoform expressed by rods has not been conclusively identified (*Lee et al., 2003*; *Feigenspan et al., 2004*). The rod–cone junctional plaques are very small, each containing few connexon channels (*Raviola and Gilula, 1973*), and although rod signals were recorded in cat (*Nelson, 1977*) and in macaque cones (*Schneeweis and Schnapf, 1995*, *1999*; *Hornstein et al., 2005*), the extent to which rod–cone coupling contributes to mammalian vision remains unclear. On the one hand, psychophysical and electroretinographic (ERG) experiments, in humans, detected putative correlates of rod–cone coupling (reviewed by *Sharpe and Stockman, 1999*), and ganglion cell and ERG recordings, in mice lacking Cx36, supported the view that rod–cone coupling is relevant for dim light vision (*Deans et al., 2002*; *Volgyi et al., 2004*; *Abd-El-Barr et al., 2009*; *Seeliger et al., 2011*); on the other hand, alternative mechanisms could explain the human data (*Sharpe and Stockman, 1999*), and recordings in cone bipolar cells in Cx36 knockout mice suggested that rod–cone coupling plays a marginal role in rod signal flow (*Pang et al., 2010*, *2012*).

Highly relevant to this debate is the fact that rod–cone GJs appear to be dynamically regulated: measurements of the extent of tracer diffusion in the outer rodent retina from the surface of a cut made with a razor blade (a technique referred to as 'cut-loading') (*Ribelayga et al., 2008*; *Ribelayga and Mangel, 2010*; *Li et al., 2013*), and ERG (*Heikkinen et al., 2011*) suggest that endogenous

\*For correspondence: lorenzo.cangiano@gmail.com

**Competing interests:** The authors declare that no competing interests exist.

**Reviewing editor**: Jeremy Nathans, Howard Hughes Medical Institute, Johns Hopkins University School of Medicine, United States

**eLife digest** People can see in a range of light levels—from dim moonlight to bright midday sun—because our eyes contain two types of light-sensitive cells: rods and cones. Rods are more plentiful than cones, and while they are sensitive at low light levels, rods can only provide grey-scale vision. Further, bright light can rapidly 'dazzle' the ability of rods to see in near-darkness, and they are slow to recover when this happens. In contrast, cones need bright light to function, but allow us to see in colour.

The signals received by rods and cones are sent through the optic nerve to the brain, where they are interpreted as vision. However, 'gap junctions' that connect the rods and cones allow for electrical and chemical 'crosstalk' between these cells, before the signals then travel along the optic nerve. Furthermore, even though it is thought that the connections between rods and cones are regulated in response to light, the body's daily rhythms and other biochemical signals, their importance for vision is not known.

Now, Asteriti et al. have taken tissue slices from the retinas at the back of mice eyes, and measured the electrical signals generated when cones are exposed to light. This revealed that the rod-cone coupling is strong enough to make the cones responsive to dim light, just like rods. Moreover, the cones also recovered slowly after being exposed to flashes of bright light. When chemical inhibitors were used to block the gap junctions, the cones stopped behaving like rods and became less sensitive to dim light.

The findings of Asteriti et al. show that rod-cone coupling is sufficient to play an important role in vision. The next challenge is to find out what this role is, and how it might be affected by different physiological conditions, including stress and injury.

neuromodulators, light, and circadian clocks influence the level of coupling (but see *Schneeweis and Schnapf, 1999*) similarly to other retinal GJs (*Lasater and Dowling, 1985*). However, cut-loading cannot discriminate between rod–rod, rod–cone and cone–cone GJs, and it does not provide information on the absolute strength of coupling, but only on its relative changes. Therefore, the possibility that the impact of coupling on vision is strongly context-dependent raises three pressing questions: (*i*) what is the maximum level of rod–cone coupling? (*ii*) how strongly are cones influenced by rod input under these conditions? (*iii*) what is the level of coupling under different physiological states? Here we investigated the first two of these questions in mouse, a mainstay of current retina research and one in which direct proof of rod input in cones is still lacking. This important gap in our knowledge is explained by the fact that mouse cones were only recently shown to be accessible for patch clamp recordings (*Cangiano et al., 2012*).

## Results

*Cangiano et al. (2012)* showed that mouse cones can be recorded with perforated patch clamp much more frequently than their numeric proportion of ~3% of photoreceptors (*Jeon et al., 1998*). We exploited this ability to dissect rod input in cones using two light stimulation protocols, a *kinetics protocol* and a *spectral protocol*, described in the 'Materials and methods' and *Figure 1*. We present data from 74 cones.

### Cones express a rod-like sensitivity to dim flashes and slow recovery after bright flashes

The kinetics protocol was delivered in rods as a control (*Figure 2A1,A2*). As expected, the first dim G flash evoked a large response, the bright$_a$ G flash evoked a saturating response consisting of a fast peak and plateau (for an analysis of the currents involved, see *Della Santina et al., 2012*), and dim G flash sensitivity recovered slowly after the saturating flash. Rod responses run down in kinetics during patch recordings (*Cangiano et al., 2012*), a process that also alters their time course of recovery from saturating flashes (*Figure 2A1*, cf. black and gray records). Thus, the kinetics protocol was also delivered in a number of rods recorded in loose seal mode, as with this technique, light response kinetics can be stable for several hours (*Figure 2A2*). The time course of the response to the kinetics protocol of rods in the initial minutes of patch recordings (n = 19, *Figure 2A1*, black records, taken before a significant

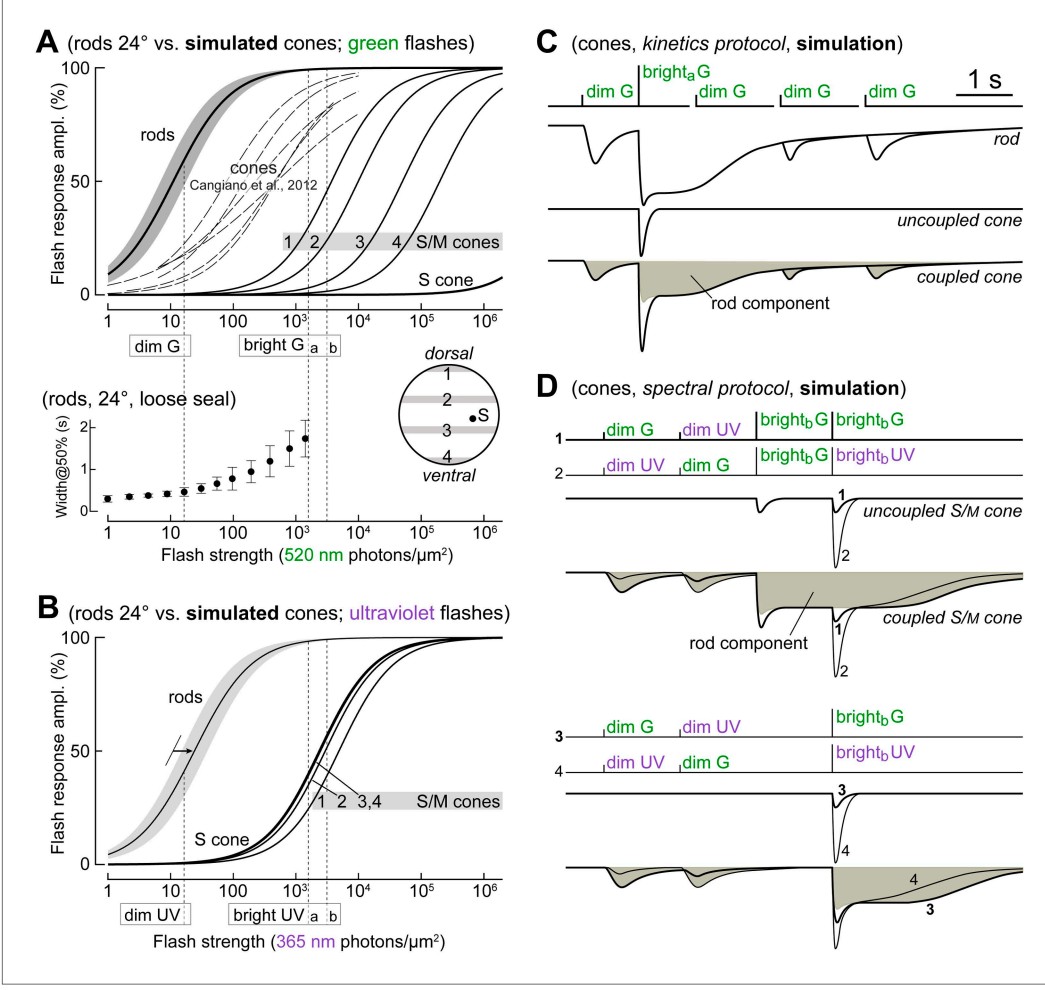

**Figure 1**. The different flash sensitivities, kinetics, and spectral preferences of rods and cones are exploited to dissect their electrical coupling. (**A** *upper graph*) Comparison of rod and cone responses to green flashes. Continuous lines on the right are simulated flash response profiles of mixed S/M cones at four retinal latitudes (1–4, from dorsalmost to ventralmost; inset in lower graph) and pure S cones, based on recently published models (***Daniele et al., 2011***; ***Wang et al., 2011***; see 'Materials and methods'). Thick line and shaded area on the left show the mean ± 1 SD of rod flash response profiles from patch and loose seal recordings in this study (n = 16; 24°C; average of Michaelis–Menten fits in individual rods). Dashed lines in the middle reproduce flash response profiles from spectrally unidentified mouse cones recently recorded with patch (reproduced from ***Figure 3***; ***Cangiano et al., 2012***, 24°C): the high sensitivity of cone photovoltages is suggestive of rod input. (**A** *lower graph*) Width at 50% amplitude (mean and SD) of rod flash responses, obtained here with loose seal recordings to avoid kinetics rundown (n = 4, 24°C). (**B**) Comparison of rod and cone responses to ultraviolet flashes. Rod response profiles in the ultraviolet were obtained by rightward shift of those in the green (panel **A**) by a factor of 2.2 (arrow; our estimate from two rods in which full flash response profiles were delivered at both wavelengths). Graphs in panels **A** and **B** were used to select dim and bright flashes for the protocols used in our experiments (**C** and **D**), aimed at dissecting rod input in cones: dim flashes (16.6 photons·μm⁻²) elicited large responses in rods while being too weak to stimulate cones, while bright flashes (a: 1570, b: 3140 ph·μm⁻²) were sufficient to saturate rods for >1 s and evoke moderate responses in cones. (**C**) *Kinetics protocol* made of sequences of three 520 nm (green, G) flashes, each consisting of dim/bright$_a$/dim flashes, with the third flash occurring at increasing delays. Expected responses in a rod, an uncoupled cone, and a coupled cone. (**D**) *Spectral protocol* made of sequences of 4 dim and bright$_b$ flashes at 520 nm (G) and 365 nm (UV). Expected responses in an uncoupled and coupled S/M-cone (UV-opsin-dominated cone). When coupled, the cone should prefer dim G to dim UV flashes, while the opposite should occur with bright flashes. The cone's intrinsic spectral phenotype is unmasked with a rod-saturating pre-flash.

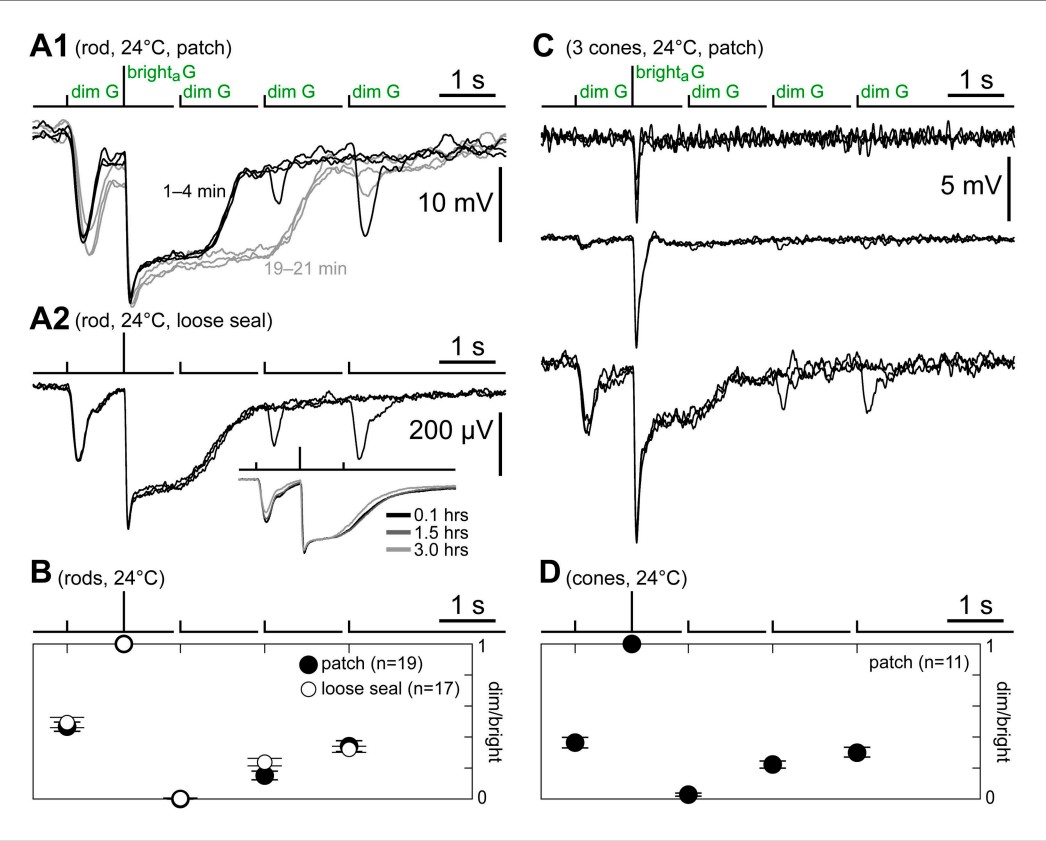

**Figure 2**. Cones express a rod-like sensitivity to dim flashes and slow recovery after bright flashes. (**A1**) Response of a patched rod to the kinetics protocol (*Figure 1C*) in the first minutes after establishing the seal (black traces). At later times, a previously described rundown of kinetics was observed (gray traces; see *Cangiano et al., 2012*). (**A2**) Loose seal recording showing a scaled version of the rod photovoltage in response to the kinetics protocol. The advantage of the loose seal approach is that no kinetics rundown takes place, even in very long recordings (inset). (**B**) Summary of rod responses to the kinetics protocol in patch (black circles; data from the first 2 min after sealing) and loose seal recordings (white circles). Dim flash responses were normalized to those of the $\text{bright}_a$ flash (bars are SEM). Rods display a large response to the first dim flash and a progressive recovery after the $\text{bright}_a$ flash. (**C**) Responses of three cones to the kinetics protocol, representing the observed spectrum of behaviors. (**D**) Summary of data from a subset of cones that displayed large dim flash responses. The time course of recovery of the dim flash response after the bright flash is comparable to that of rods. In panels **A1**, **A2**, and **C**, baselines were aligned to each other (max shift 2 mV), and in all records (except **C**/top), each trace was the average of several sweeps.

amount of rundown occurred) was similar to that in loose seal recordings (n = 17), as shown in a graph of the normalized dim flash response amplitudes (*Figure 2B*). In cones, the kinetics protocol evoked a spectrum of response types depending on the cone and on the time from seal (see below). Some cones responded only to the $\text{bright}_a$ G flash (*Figure 2C*, top), the expected behavior for uncoupled cones (*Figure 1*). However, the large majority of cones displayed, similar to rods, responses to the first dim G flash, a plateau after the $\text{bright}_a$ G flash, and a slow recovery of the dim flash response (*Figure 2C*, middle/bottom). This similarity in the time course of recovery after the bright flash emerges from comparing graphs of normalized dim flash response amplitudes during the kinetics protocol in rods (*Figure 2B*) and cones (*Figure 2D*, n = 11). This is strong evidence for the presence of rod input in cones.

## Cones shift toward a rod-like phenotype during recording

We observed in cones a progressive increase in dim and bright flash peak response amplitudes and in the plateau. The net change in their response to the kinetics protocol was a scaled version of the typical rod response (*Figure 3A*). The rapid time course of this process implied that it began with the recording itself. It did not depend on the repeated exposure of the photoreceptor to light, since it occurred also

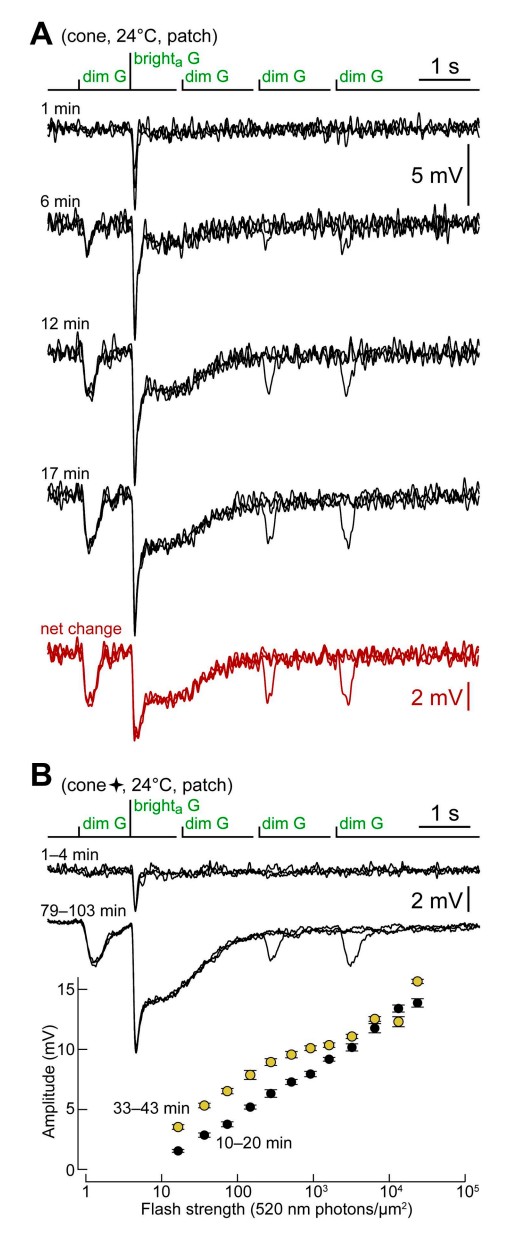

**Figure 3**. Cones shift toward a rod-like phenotype during recording. (**A**) Response of a cone to the kinetics protocol delivered at 1, 6, 12 and 17 min after obtaining the seal (records are not averages; $V_{dark} \approx -44$ mV). In this experiment, darkness was maintained between recordings. The net change between the first and the average of the last two records (12 and 17 min) matches the response of rods (**Figure 2A**). (**B**) A different cone in which the kinetics protocol was delivered at close intervals for an extended time. Records above compare the average responses of this cone to the kinetics protocol at the beginning of the experiment, with those after >1 hr. The graph below shows the selective increase in sensitivity to dimmer flashes that occurred during the recording (bars are SEM). This cone also appears in **Figure 4** (four-pointed star).

when single deliveries of the kinetics protocol were separated by several minutes of uninterrupted darkness (**Figure 3A**). A response amplitude vs flash strength graph, obtained in two time ranges in the same cone (**Figure 3B**), highlights how cones acquired light sensitivity at intensities normally covered by rods. Note that collecting each full flash response curve of **Figure 3B** required ~10 min, a time comparable to the time course of the spontaneous increase; therefore, different flash strengths were unavoidably delivered at different levels of progress of the phenomenon. **Figure 4** shows, in simplified form, the evolution of dim G flash response amplitude in 50 cones. Its rate of increase varied greatly among cones. When possible we estimated the dim G flash response amplitude prior to patching, by extrapolating to time zero the values observed in the first minutes of recording (**Figure 4**, short red horizontal bars). Based on these estimates, it appears that most cones responded to dim G flashes already before patching, although response amplitudes were generally modest when compared to those expressed during the recordings. If the dim flash sensitivity of cones is due to rod input, then these observations imply that rod–cone junctional conductance is not hardwired, but it is modifiable in a wide range.

## Rods and cones recover from a rod-saturating background at the same rate

We attempted to isolate the pure cone component by saturating rods with a continuous light background of 6100 photons·$\mu$m$^{-2}$·s$^{-1}$ at 520 nm (for 5 min) based on *ex vivo* ERG data in mouse (*Heikkinen et al., 2011*) and our own rod data obtained with patch clamp and in loose seal. Control recordings confirmed that this particular background was, indeed, rod saturating (**Figure 5A1**) and showed that, upon returning to darkness, rod responses to the kinetics protocol recovered to near control levels after ~2 min (n = 3). This is presented in **Figure 5A2**, where dim G flash, bright$_a$ G flash, and slow plateau response amplitudes are plotted normalizing them to their control values before exposure to the background. The same background was delivered in cones expressing prominent dim G flash responses. In contrast to rods, a sharp response to the bright$_a$ G flash persisted during the background, while the slow plateau disappeared as one would expect if it was generated by rods (**Figure 5B1**). The sharp peak did not represent a residual rod response, since increasing flash strength led to a marked increase in its amplitude (**Figure 5B1**,

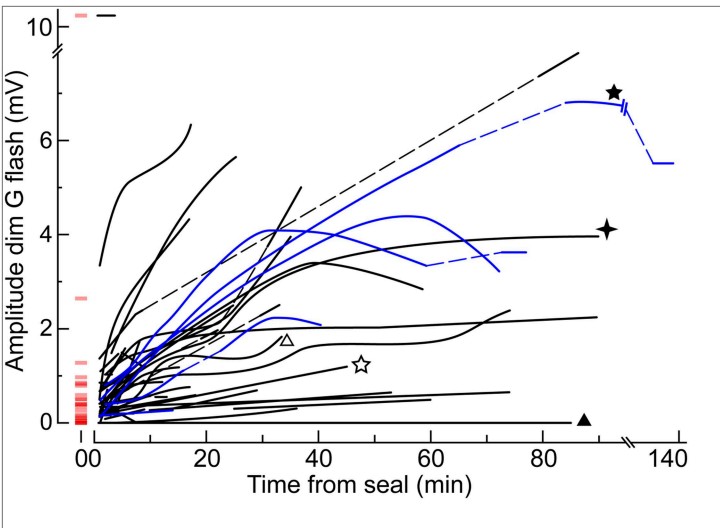

**Figure 4**. The dim flash sensitivity of cones increases during recording. Time course of the peak amplitude of the response to the dim G flash in 50 cones. Each curve corresponds to a different cell and is a qualitative fit to the raw data points. In a subset of cones, the unperturbed amplitude of the dim G flash response could be extrapolated with a reasonable degree of confidence (red horizontal segments at time zero). Some cones were recorded with an *EGTA zero Ca²⁺ perforated patch* solution in the pipette (blue lines; see 'Results'). Dashed lines represent gaps in the data resulting from the delivery of other protocols. Some cones also appear in other figures (stars and triangles).

arrow). Importantly, when darkness was restored, the dim and bright flash responses recovered to near control levels in ~2 min (n = 3), in agreement with what was observed in rods (cf. *Figure 5 B2* with *A2*). This strongly suggests that the peculiar features displayed by cones—a high light sensitivity and slow kinetics—may be entirely explained by rod input fed into the cone pathway through GJs.

*Heikkinen et al. (2011)* interpreted the effects of non rod-saturating light backgrounds on the cone-driven ERG flash response as evidence that light uncouples rod–cone GJs, with a time course of minutes. Given our observation that the rod component in cones recovers from a prolonged rod-saturating background with a similar time course as the rods themselves, we could not confirm the conclusions of that study. It is possible, however, that the process leading to a progressive increase in rod–cone coupling during patch recordings might interfere with the relevant signaling pathways.

## Irrespective of their dominant cone opsin, for dim flashes, cones respond more to green light

In the presence of rod input, one should observe a shift in the spectral preference of S- and S/M-cones toward that of rods. The spectral protocol (*Figure 1D*) enabled us to rapidly determine the apparent spectral preference of cones for dim and bright_b flashes, as well as their intrinsic spectral preference by removing any rod contribution with a rod-saturating pre-flash. We delivered the spectral protocol in rods and cones and quantified spectral preference by the ratio of the peak response amplitudes to G over UV light (for the same flash strength in photons·μm⁻²). As expected, rods were more sensitive to dim G than to dim UV flashes (*Figure 6A1,A2*, arrows; same rod), with an estimated dim G/UV ratio of 2.7 (SEM 0.2; n = 8) (all rhodopsins have a prominent secondary absorption peak in the ultraviolet [*Rodieck, 1973*], including mouse rhodopsin [*Lyubarsky et al., 1999*]). For both G and UV bright_b flashes, rods expressed saturating responses (ratio of 1; *Figure 6A2*, filled box), while they did not respond to bright_b flashes delivered after a rod-saturating pre-flash (*Figure 6A1*, empty box). *Figure 6A1,A2* shows a cone exposed to the same spectral protocol delivered in rods. Surprisingly, while for dim flashes we observed a larger response to G than to UV light (*Figure 6A1,A2*, arrow-heads; same cone), the response to bright_b flashes delivered after the rod-saturating pre-flash showed that this cone had an intrinsic UV preference (*Figure 6A1*, empty circle) and was therefore a mixed S/M cone. We found that all cones stimulated with dim flashes displayed larger responses

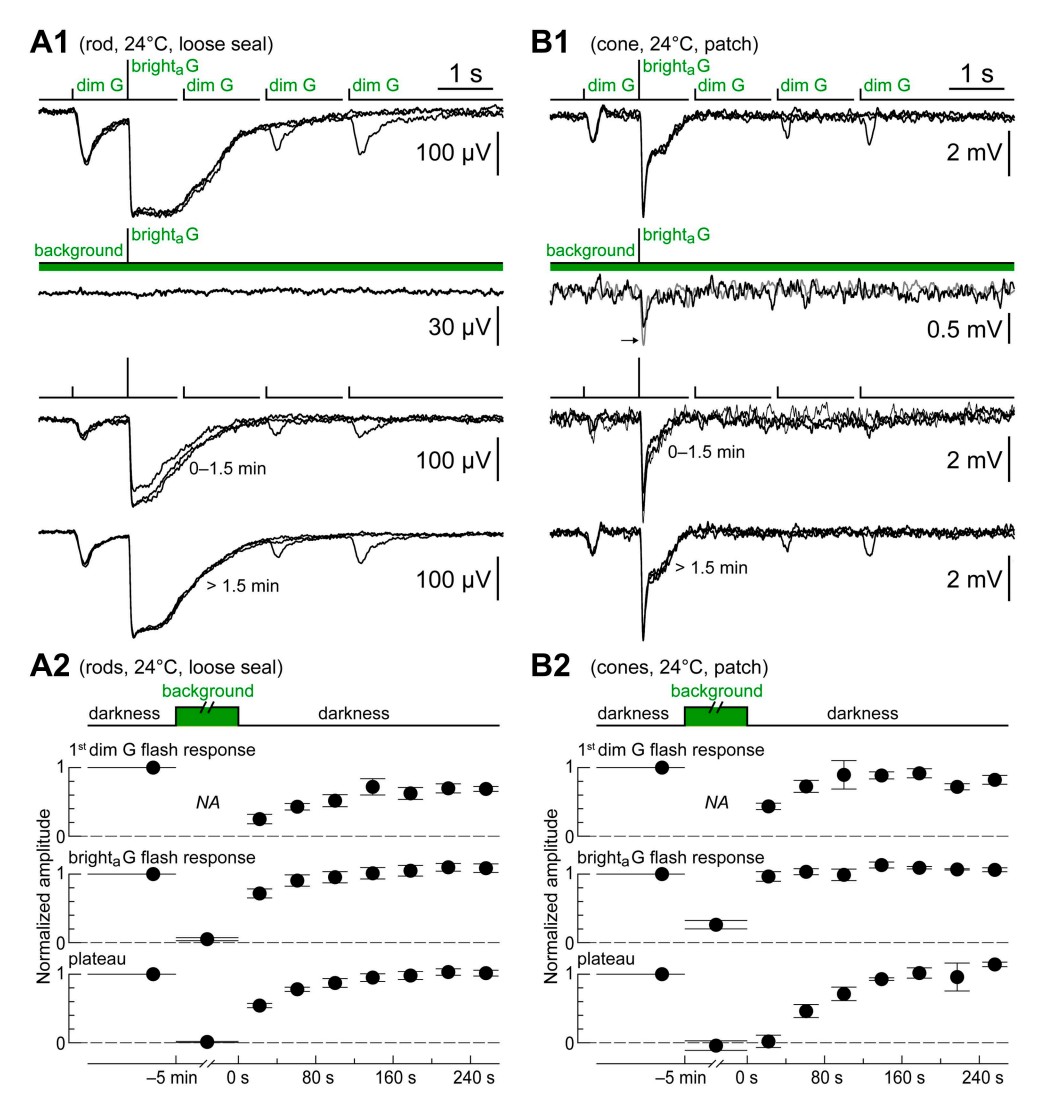

**Figure 5**. Rods and cones recover from a rod-saturating background at the same rate. (**A1**) Loose seal recording of a rod showing the complete suppression of its response to the kinetics protocol by a saturating light background (520 nm, 6100 photons·μm$^{-2}$·s$^{-1}$ for 5 min; dim G flashes were not delivered during the background), and its recovery upon returning to darkness. (**A2**) Graphs summarizing the effect of the light background on this and two additional rods. The response amplitudes of the first dim G flash, the bright$_a$ flash, and the plateau (0.4–0.6 s post bright flash) were normalized to their control values (bars show the SEM, while their horizontal extent shows the time range of the underlying flashes). (**B1**) Recording of a cone of high light sensitivity and slow kinetics showing the effect of the same rod-saturating background. In contrast to rods, a fast response component persisted in the cone during the background (arrow; gray trace shows the effect of increasing flash strength by a factor of 3.7). (**B2**) Graphs summarizing the effect of the light background on this and two additional cones. The time course of recovery in cones matched that of rods. All records in **A1** and **B1** are averages obtained in the specified time ranges.

to G than to UV light (n = 20). This behavior was independent of their intrinsic spectral preference in a wide range of G/UV ratios (*Figure 6B*) (corresponding to a wide range of M- vs S-opsin expression levels), again supporting our hypothesis that dim flash responses in cones are driven by rods.

An important question is whether the cones' maximal strength of coupling to rods varies with their dominant opsin type. We examined this by plotting the maximum dim G flash response amplitude

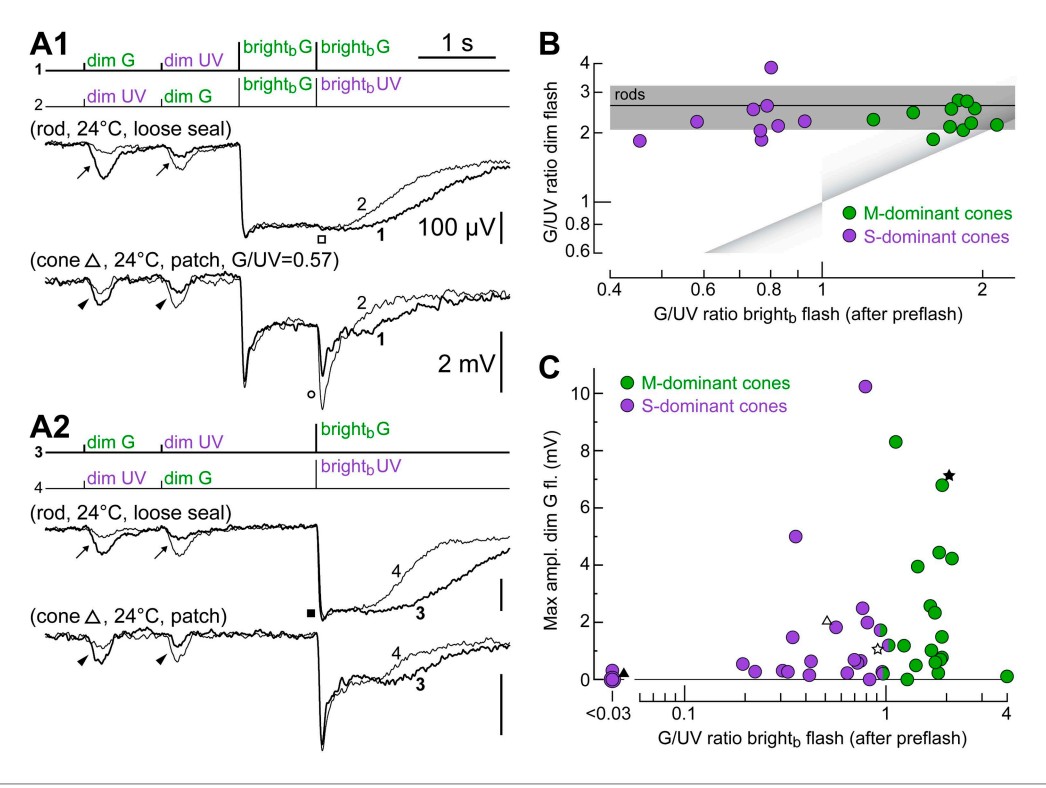

**Figure 6**. Irrespective of their dominant cone opsin, for dim flashes, cones prefer green light. (**A1** and **A2**) Responses of a rod and a cone to the spectral protocol (*Figure 1D*). As expected, the rod was more sensitive to dim G than dim UV flashes (arrows), it saturated with bright$_b$ G and bright$_b$ UV flashes (filled box) and did not respond after a rod-saturating pre-flash (empty box). Similar to the rod, the cone was more sensitive to dim G flashes (arrowheads). However, its responses after a rod-saturating pre-flash unmasked an intrinsic preference for UV light (empty circle). (**B**) Cones of widely varying intrinsic spectral preference (corresponding to widely varying M- vs S-opsin expression levels) display a rod-like preference for green light when tested with dim flashes. Spectral preference was quantified as the ratio of G and UV flash response amplitudes. Line and shaded areas show the mean dim flash preference ± 1 SD in rods (four loose seal and four patch recordings). Triangular shades show the expected location in the graph of uncoupled cones. (**C**) Both M- and S-dominant cones couple to rods, as shown by a plot of the maximum dim G flash response amplitude observed in each cone vs its intrinsic spectral preference.

observed in each cone vs its intrinsic spectral preference (*Figure 6C*). The graph shows that both M- and S-dominant cones could possess or acquire dim flash sensitivity during recordings. Performing a finer analysis under our experimental conditions would be complicated by the fact that the maximum level of coupling displayed by cones is strongly dependent upon recording duration, which varied widely. We also recorded from three putatively pure S-cones (i.e. *blue* cones; *Figure 6C*, leftmost circles), an important sample since the immunohistochemically derived frequency of these neurons is very low (*Haverkamp et al., 2005*). In one of these cones (*Figures 4, 6C, 7C*, black triangle), the seal was maintained for >85 min, but no coupling was detected. The remaining two cones were recorded for only 4 and 7 min and thus (*i*) little averaging of their responses to the spectral protocol could be performed to improve signal over noise, and (*ii*) the possible development of coupling could not be monitored. With these limitations in mind, in one cone, no coupling could be detected, while in the other, a dim G flash response of ~0.3 mV may have been present although superimposed on a noisy baseline.

*Figure 7* also serves the purpose of illustrating in greater detail three cones shown in previous plots (*Figure 4* and *Figure 6C*, five-pointed stars and black triangle) exhibiting different relative opsin expression levels: an M-dominant cone initially uncoupled, which then develops strong coupling (panel A); a weakly coupled S/M cone, with approximately equal sensitivity to G and UV light

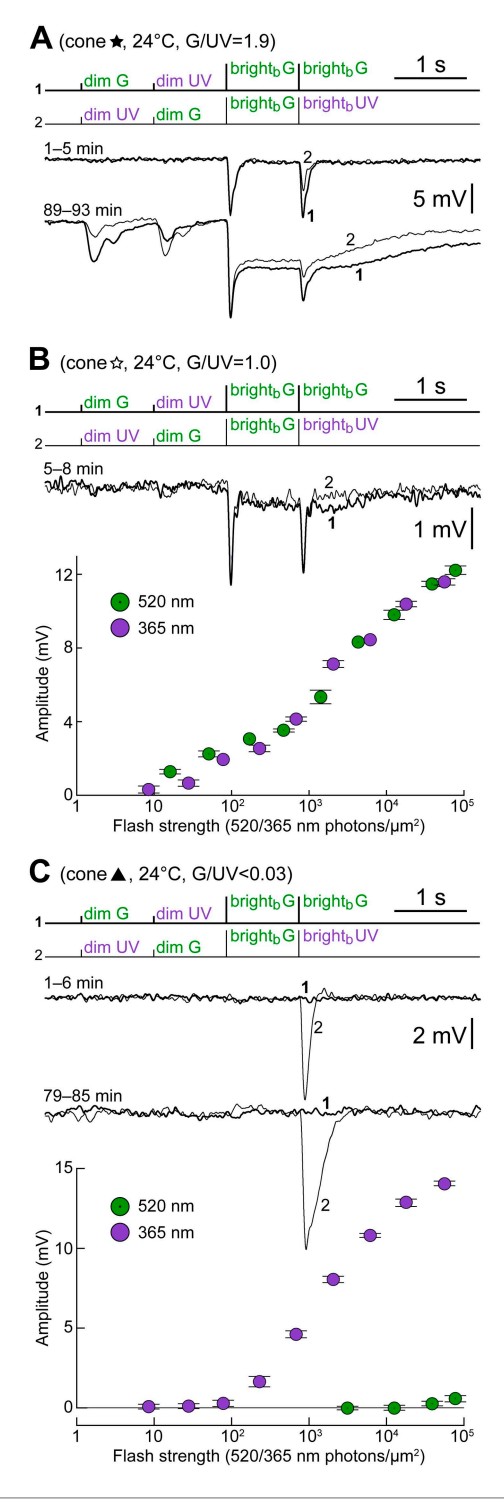

**Figure 7**. Examples of cones with different S- vs M-opsin expression levels. (**A**) Initially uncoupled M-dominant cone, which then develops strong coupling (also shown in **Figure 4** and **Figure 6C**, labeled by a black five-pointed star). (**B**) Weakly coupled S/M cone with approximately equal sensitivity to G and UV light (white five-pointed star in previous figures). (**C**)
*Figure 7. Continued on next page*

(panel B); an uncoupled presumably pure S-cone (panel C).

## Dim and bright flash responses originate from separate electrotonic compartments

If, as we hypothesized, the cone dim flash response originates in coupled rods, while the bright flash response originates in both photoreceptors, they should have a different *apparent* reversal potential (assuming the actual reversal potential $E_{OS}$ in rods and cones to be the same; $E_{OS} \approx 0$ mV; *Luo et al., 2008*). The reason is that the cone membrane potential at the pipette tip ($V_{Cone}$) would differ from the membrane potential of its coupled rods ($V_{Rod}$) when a current flows through the finite resistance of the GJs ($i_{GJ}$), a condition occurring when the cone is held depolarized via the pipette (*Figure 8A*, equivalent circuit). In essence, any coupled rods would represent electrotonically distant compartments from the patch pipette. The prediction is that there should be a range of depolarized cone membrane potentials (beyond $E_{OS}$) within which $V_{Rod(DARK)} < E_{OS}$ *and* dim and bright flash responses display opposite polarities (*Figure 8A*, expected responses: bottom row) (note that, since homotypic Cx36 GJs decrease their conductance in the presence of large voltage differentials (*Bukauskas, 2012*), it is entirely possible that all one would observe when $V_{Cone} > E_{OS}$ is an uncoupling, that is disappearance of dim flash responses). If, on the other hand, dim and bright flash responses originated entirely in cones, their polarity should always be concordant.

A full analysis of the voltage-dependence of the photoresponses could not be carried out due to the instability of recording at positive membrane potentials and the need of acquiring several responses for averaging. We chose instead a more limited approach of delivering the spectral protocol in cones sensitive to dim flashes, while they are depolarized well beyond the reversal potential of their photoresponse. This was done both in current clamp (CC) and voltage clamp (VC). In CC, we imposed $V_{Cone(DARK)} > E_{OS}$ by constant current injection. In all five cones tested, we observed that the bright$_b$ flash responses reversed in polarity (i.e., depolarizing) and actually became larger than in control conditions. In contrast, four of these cones had their dim flash responses reduced in amplitude to the point of being no longer distinguishable from noise, while the remaining one retained a small hyperpolarizing response (*Figure 8B*). This result suggests that in these conditions, $V_{Rod(DARK)}$ was slightly below or anyway near $E_{OS}$ (*Figure 8A*, middle and bottom

*Figure 7. Continued*

Uncoupled, presumably pure S-cone (black triangle in previous figures). All records are averages obtained in the specified time ranges. Bars are SEM.

rows). In VC, the photocurrent responses of a cone to the spectral protocol were recorded at holding potentials ($V_{Cone(HOLD)}$) of −40 mV and +60 mV. At the depolarized potential, bright flash photoresponses were reversed and of much larger amplitude, while the dim flash responses became undetectable (*Figure 8C*). This suggests that also in this case (VC recording) $V_{Rod(DARK)}$ was close to $E_{OS}$ (*Figure 8A*, middle row). Again, it is possible that the absence of dim flash responses was partly due to a voltage-dependent reduction of the junctional conductance.

## Blocking GJs reverts cones to their intrinsic phenotype

As a further confirmation of the results obtained thus far, we tested the specific GJ blocker meclofenamic acid (MFA), previously known to be effective at GJs containing Cx36 in AII amacrine cells (*Pan et al., 2007*; *Veruki and Hartveit, 2009*). In three control rod recordings, 100 µM MFA was superfused with essentially no effect, except for a marginal reduction in light sensitivity that we attribute to the normal rundown observed during long rod recordings. In four cones tested, on the other hand, superfusion with 100 µM MFA markedly reduced both dim flash responses and the slow plateau after bright flashes. An example is shown in *Figure 9A1*, where a prominent level of coupling was reached about 30 min from seal formation, after which MFA was delivered. The slow pharmacodynamics of the blocker observed in our recordings (*Figure 9A1*, lower plot) confirmed a previous report in AII amacrine cells (*Veruki and Hartveit, 2009*). The significant rundown of the cone response kinetics, particularly evident toward the end of the experiment, was first described in *Cangiano et al. (2012)* and is unrelated to the effect of the blocker, since fast cone responses were observed when the seal was made with MFA already present in the bath (n = 2). Comparing graphs of response amplitude vs flash strength before and during perfusion with MFA in the same cone highlights how junctional coupling transforms the sensitivity profile of the cone making it responsive to dim flashes, thereby widening its dynamic range (*Figure 9A2*). Although control data for this graph were obtained before the cone reached its full coupling potential, a remarkable effect of the blocker on dim flash sensitivity was already visible.

Uncoupled cones, including those recorded during superfusion with MFA (*Figure 9—figure supplement 1*), displayed markedly decreased flash sensitivities compared to those found in *Cangiano et al. (2012)*, and approached those predicted using published estimates from functionally rodless retinas (*Figure 1A,B*, simulated curves on the right; 'Materials and methods').

## Rod–cone coupling is also expressed near body temperature

As the experiments shown so far were performed near room temperature, we verified the occurrence of rod–cone coupling also near body temperature. The response to the kinetics protocol of a rod recorded in loose seal (*Figure 10A*) showed, as expected, faster kinetics near body temperature (cf. *Figure 2A2*). Nevertheless, the rod was still unable to respond to the dim G flash delivered at the earliest delay following the bright$_a$ G flash. In all five cones recorded near body temperature, we observed dim G flash responses (0.28, 0.32, 0.43, 1.47, and 1.70 mV) and plateaus after bright$_a$ G flashes (the response of 1.70 mV is shown in *Figure 10B*). In one of these cones, we tested the effect of the GJ blocker MFA (100 µM): both dim flash responses and plateaus following the bright$_a$ flash were abolished, leaving behind a fast response to the bright$_a$ flash (*Figure 10B*). The flash response profile of this cone, acquired during superfusion with MFA (*Figure 10B*), revealed that it was significantly more sensitive to G light than the dorsalmost (i.e., the 'greenest') simulated cones in *Figure 1A*. This was not necessarily unexpected, given that the simulated profiles: (*i*) are based on models which predict the *average* ratio of S- to M-opsin expression over a large cone population, and (*ii*) assume that all cones express the same amount of opsin ('Materials and methods').

## The spontaneous increase in coupling does not require Ca²⁺ changes in the cone

The progressive increase in rod–cone coupling observed in our recordings was not evoked by slicing of the retina but rather by some interaction between the recording pipette and the cone and/or its nearby rods: cones were patched not earlier than 1 hr after slicing and in some cases, after several hours,

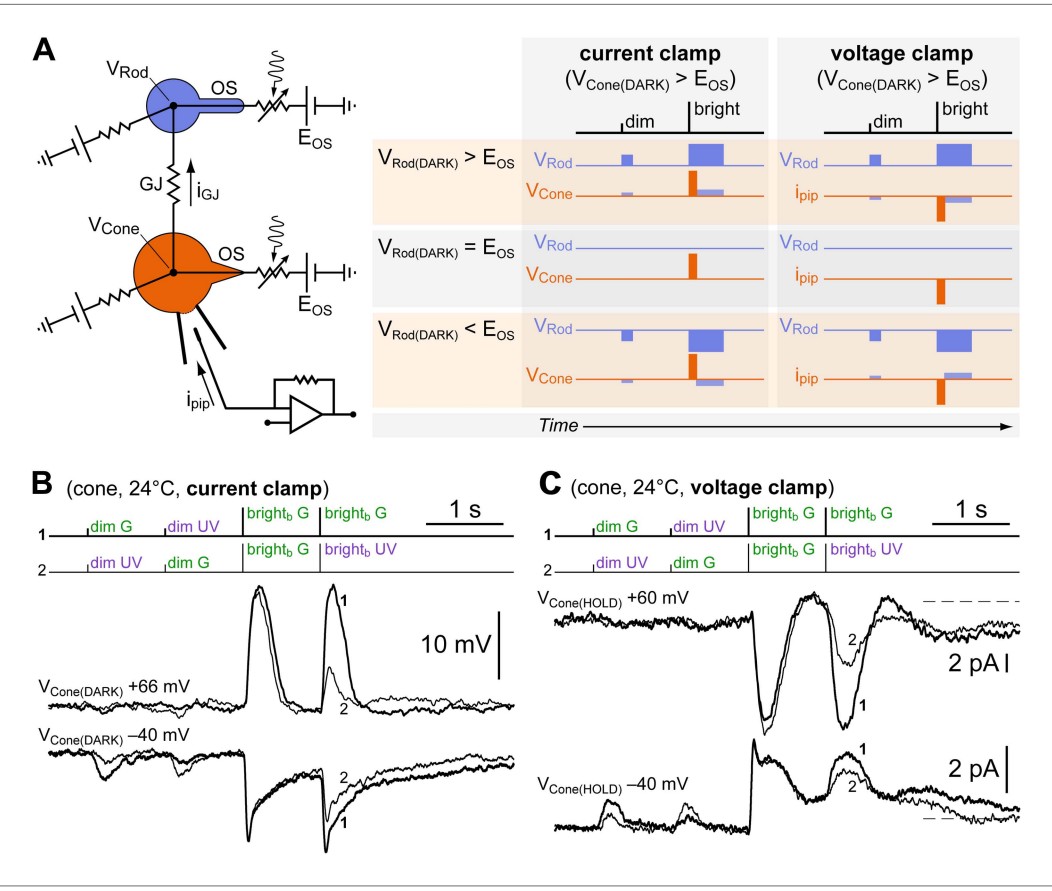

**Figure 8**. Dim and bright flash responses in cones originate from separate electrotonic compartments. (**A** *left*) Equivalent circuit of a recorded cone (orange) coupled to neighboring rods (blue) via GJs (OS, outer segment; $E_{OS}$, reversal potential of the rod/cone light sensitive conductance; GJ, gap junction; $i_{GJ}$, junctional current; $i_{pip}$, pipette current; $V_{Rod}$ and $V_{Cone}$, rod/cone membrane potential). (**A** *right*) When the cone is depolarized, either in current clamp (by constant current injection) such that $V_{Cone(DARK)} > E_{OS}$, or in voltage clamp such that $V_{Cone(HOLD)} > E_{OS}$, a junctional current $i_{GJ}$ will flow into the rods and depolarize them beyond, at, or below $E_{OS}$. Each of these three possible outcomes is expected to lead to the indicated different combinations of response polarities when delivering dim and bright flashes. (**B**) A reduced version of the spectral protocol (sequences 1–2 in *Figure 1D*) was delivered with a cone recorded in current clamp in control conditions ($V_{Cone(DARK)} = -40$ mV) or during depolarization by constant current injection beyond the reversal potential of its light-sensitive conductance ($V_{Cone(DARK)} = +66$ mV). While bright$_b$ flash responses reversed polarity, dim flash responses became smaller but did not reverse. This is *not* compatible with an origin of the dim and bright flash responses in the same electrotonic compartment, and matches one of the predicted outcomes for coupled cones (panel **A**, $V_{Rod(DARK)} < E_{OS}$). The moderate shift toward G in the spectral preference displayed by this cone (second bright$_b$ flash) could be explained by cone–cone coupling and/or by initial recovery from saturation of its coupled rods (see 'Discussion'). (**C**) The same experiment as in panel **B** but performed in voltage clamp in a different cone ($V_{Cone(HOLD)} = -40$ mV and $+60$ mV). Dim flash responses could not be detected above noise when the cone was depolarized, despite the presence of large inverted bright$_b$ flash responses. This matches a different predicted outcome for coupled cones (panel **A**, $V_{Rod(DARK)} = E_{OS}$). The slight shift toward G in the spectral preference displayed by this cone (second bright$_b$ flash) is likely explained by a slow 'bump' in the plateau displayed after bright$_b$ flashes, present only at $-40$ mV (not shown, but observed in sequence three of the spectral protocol). The outcome predicted for the case of $V_{Rod(DARK)} > E_{OS}$ in panel **A** was never observed. Records are averages of 3–4 sweeps.

yet in most cones, initial coupling upon sealing was weak, developing rapidly thereafter (*Figure 4*). To shed light on the nature of this interaction, we examined whether the mechanism underlying the coupling process in photoreceptors is related to phenomenologically similar ones found in other systems expressing Cx36 (*Zoidl et al., 2002*; *Del Corsso et al., 2012*; *Veruki et al., 2008*). Two possible drivers of the spontaneous coupling increase emerging from these studies were (*i*) dialysis of the cell by the

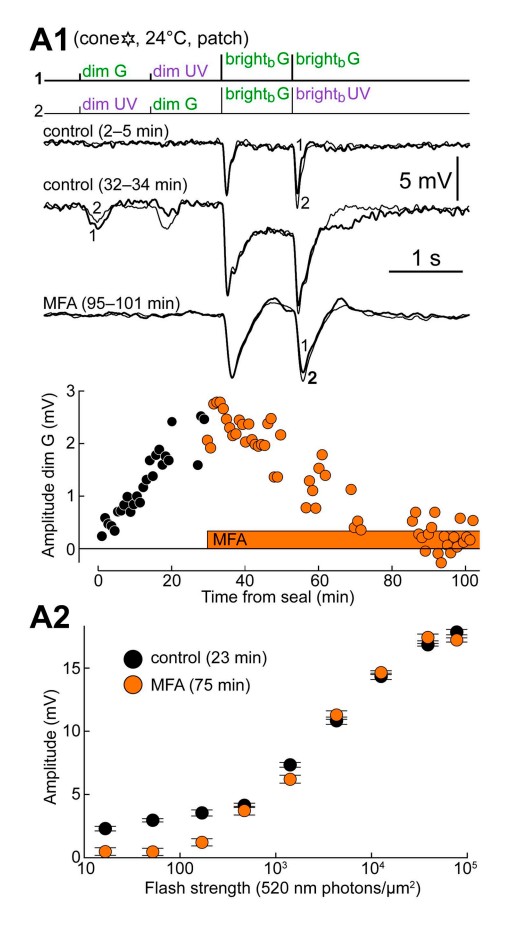

**Figure 9**. Blocking gap junctions reverts cones to their intrinsic phenotype. (**A1**) Response of a cone to the spectral protocol during its spontaneous shift toward a rod phenotype (top and middle records) and subsequent superfusion with the GJ blocker meclofenamic acid (MFA, 100 μM; bottom records). MFA abolished both dim flash responses and bright flash plateaus. Records are averages. (**A2**) Response amplitude vs flash strength for the same cone as in **A1** before and during perfusion with MFA. Note the selective reduction in dim flash sensitivity. Bars are SEM.

The following figure supplements are available for figure 9:

**Figure supplement 1**. Uncoupled cones display light sensitivities comparable to those predicted from literature (compare with *Figure 1A,B*, simulated curves).

**Figure supplement 2**. The spontaneous increase in coupling and MFA had a limited impact on our estimates of intrinsic cone spectral preferences.

pipette, and (*ii*) an increase in intracellular calcium, both occurring as a consequence of *whole cell* recordings ('Discussion'). However, the fact that we observed this phenomenon in perforated patch clamp recordings, a technique known to cause minimal disturbance to the intracellular environment, makes it unlikely that in mouse cones, it is caused by pipette-cell dialysis (we could exclude an unintended rupture of the patch membrane, since the amphotericin-B present in our pipette solution would have perforated the cell membrane, depolarizing it close to 0 mV and shunting the photovoltages—effects that we indeed observed when going whole cell at the end of recordings to stain the neurons with Lucifer Yellow, LY). Nonetheless, we examined the remote possibility that amphotericin-B (nominally 400 μM) formed $Ca^{2+}$-permeable pores (*Romero et al., 2009*) and that the diffusion of calcium ions from the pipette into the cone led to an opening of GJs ($Ca^{2+}$ traces are normally present in the high purity water used for intracellular solutions). In four of five cones recorded with 1 mM EGTA in the pipette (*EGTA zero $Ca^{2+}$ perforated patch* solution), we observed the same progressive increase in coupling (*Figure 4*, blue lines).

These results effectively rule out an involvement of pipette $Ca^{2+}$ but leave open the possibility that an increase in free $[Ca^{2+}]_i$ in the cone, promoted by the pipette, drives the coupling process. We thus performed whole cell recordings (i.e., without amphotericin-B) with the goal of dialyzing cones with a low $Ca^{2+}$ buffered solution. Seals were made on cone pedicles (as confirmed by LY staining, *Figure 11E*) to ensure a rapid and effective '$Ca^{2+}$ clamp' on the cone side of the junctional contacts (located on short telodendria that protrude from the pedicles; *Tsukamoto et al., 2001*; *O'Brien et al., 2012*). This represented an important step, given what is known about the active compartmentalization of free $[Ca^{2+}]_i$ in cones (*Wei et al., 2012*). We recorded from three cones with an *EGTA low $Ca^{2+}$ whole cell* solution (50 nM free $[Ca^{2+}]_i$; see 'Materials and methods'), of which only one had weak coupling to rods upon breaking the patch. In two of the three cones, rod coupling increased rapidly in the first minutes of recording (*Figure 11A*), while in the third cone, coupling did not develop despite a stable recording for more than 1 hr (not shown). Further, we recorded from three cones with a *BAPTA low $Ca^{2+}$ whole cell* solution (50 nM free $[Ca^{2+}]_i$) to rule out an involvement of fast and localized calcium transients. While all three cones were initially uncoupled, they rapidly increased their coupling to rods up to moderate levels (*Figure 11B*). Since in all these experiments free calcium in the pedicle was buffered at very low levels compared to those normally

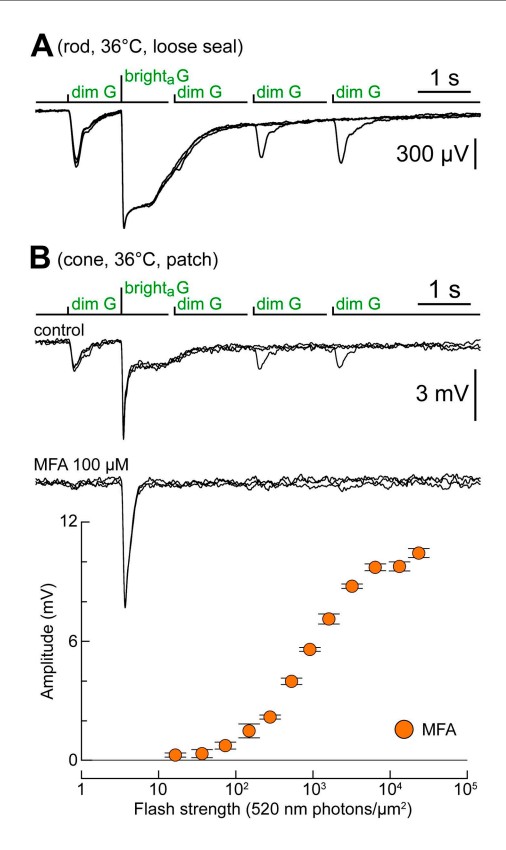

**Figure 10**. Rod–cone coupling is also expressed near body temperature. (**A**) Loose seal recordings at 36°C in a rod, showing its response to the kinetics protocol. As expected, rod recovery from the bright$_a$ flashes was faster compared to near room temperature. (**B**) Cone recorded at 36°C displaying a rod-like phenotype in response to the kinetics protocol (upper records). MFA abolished dim flash responses and slow plateaus after the bright flash (lower records). The graph shows the G flash response profile of the cone during superfusion of MFA.

present in darkness (~2 μM spatially averaged: *Szikra and Krizaj, 2006*), we conclude that high calcium (on the side of the cone) is not required for the expression of the progressive increase in coupling. High calcium, however, could favor coupling. To test this, we recorded from six cones with an *EGTA high Ca$^{2+}$ whole cell* solution (approx. 5 μM free [Ca$^{2+}$]$_i$), all uncoupled or weakly coupled upon breaking the patch. Among five cones that we could monitor for sufficient time, one developed strong coupling over several minutes (*Figure 11C*), while the remaining four displayed weak coupling, which, however, did not appear to increase over time. Thus, no obvious favoring effect on coupling of high calcium emerged from this group of cones.

*Figure 11D* summarizes the time course of coupling for the cones shown in panels A–C, by plotting the dim G flash response amplitude. Also shown is the time course of the amplitude of bright$_b$ G flash responses delivered after the pre-flash. Note that the intrinsic responses of these cones ran down rapidly (cf. lower records in panels A–C, arrows). This run down occurred in all low [Ca$^{2+}$]$_i$ and in two high [Ca$^{2+}$]$_i$ recordings. Interestingly, in the cones resistant to rundown, the distal portion of the neuron (cell body and outer segment) was only weakly stained by LY, as if a barrier to diffusion was present in the axon. Taken together, these experiments appear to exclude any major contribution in the coupling increase of Ca$^{2+}$, at least on the cone side of the GJs.

## Discussion

Patch clamp recordings in the mouse retina, a key model in current vision research, were combined with light stimulation, current injection, and pharmacology, to show that rod–cone coupling may have a remarkable impact on cone function. The possibility that what we interpret as rod input could instead represent a change in cone phototransduction can be rejected, as it is contradicted by our findings that: (*i*) the GJ blocker MFA abolished rod-like responses in cones, (*ii*) dim and bright flash responses have different reversal potentials, (*iii*) in many cones, dim and bright flash responses (delivered after a rod-saturating pre-flash) display opposite spectral preferences. Moreover, in preliminary recordings from the cones of mice lacking Cx36, we did not observe rod-like features (*Asteriti et al., 2013*).

### Cone opsin expression gradients and rod–cone coupling

Except for a small number of pure S-cones (i.e. *blue* cones), most mouse cones reportedly express both S and M opsins, with their expression ratio varying along the dorsoventral axis of the retina (*Applebury et al., 2000*; *Haverkamp et al., 2005*; *Daniele et al., 2011*; *Wang et al., 2011*). Moreover, any pure M-cones would respond to UV light since M-opsin has a prominent secondary absorption peak in the near UV region (the β-band; *Govardovskii et al., 2000*). It was thus not surprising that almost all of our recorded cones were intrinsically sensitive to both G and UV light (*Figure 6* and *Figure 7*, cf. with *Figure 1A,B*). Our dissection and recording techniques did not allow us to identify their dorsoventral position in the retina, so that we could not directly examine possible correlations between their location

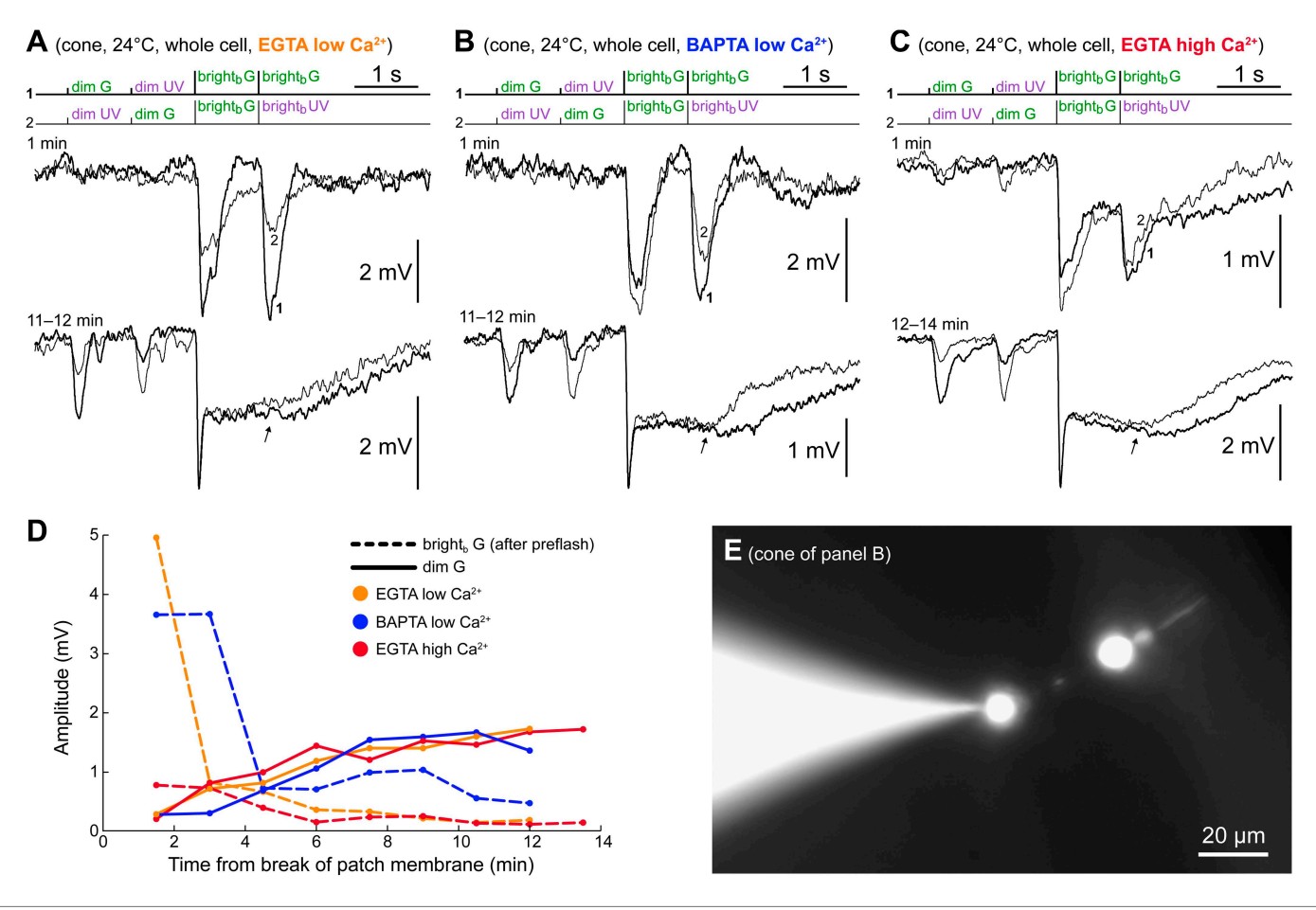

**Figure 11**. The spontaneous increase in coupling does not require $Ca^{2+}$ changes in the cone. (**A–C**) Three cones recorded in whole cell patch clamp using either low $Ca^{2+}$ intracellular solutions (50 nM free $[Ca^{2+}]$) buffered with EGTA (panel **A**) or BAPTA (panel **B**), or a high $Ca^{2+}$ intracellular solution (approx. 5 μM free $[Ca^{2+}]$) buffered with EGTA. All three cones expressed the same time-dependent increase in coupling to rods observed in perforated patch recordings: increased dim flash response and bright flash plateau amplitudes. Note that, in contrast to the perforated patch, the component of the light response originating in the cones themselves ran down rapidly (second $bright_b$ flashes, arrows). Records are averages of 1–3 sweeps. (**D**) Time course of rod (continuous lines) and cone (dashed lines) response components in the experiments shown in panels **A–C**. (**E**) Pedicles (cone synaptic terminals) were targeted in all of these whole cell recordings to ensure a rapid and effective calcium clamp at the GJs, which are located on the adjacent telodendria. The image shows a Lucifer Yellow stain of the cone in panel **B** with the pipette sealed on the pedicle (final image obtained by blending two photographs acquired on slightly different focal planes; pedicle and cell body appear larger than their actual size due to an intentional overexposure during acquisition, implemented to highlight the dim outer segment).

and the level of maximal coupling. We could however conclude that both S-dominant cones (known to be mid and ventrally located) and M-dominant cones (dorsally located) are able to couple strongly to rods (*Figure 6C*). As for pure S-cones, our small recorded sample is in no way sufficient to conclude that they have a reduced propensity to couple to rods, since we found uncoupled cones also within the S/M group. Interestingly, there is some evidence in macaque that blue cones form fewer junctional contacts with rods compared to other cone types (*O'Brien et al., 2012*).

It is important to note that the G/UV ratios obtained in our sample of cones after rod-saturating pre-flashes (*Figure 6C*) *do not* provide a completely unbiased representation of the relative sensitivity to G over UV light within the native cone population. First, while we recorded from both central and peripheral retina, it is unlikely that we obtained a uniform sampling at all eccentricities; this, combined with the dorsoventral gradients in S- and M-opsin expression (see references above), will distort the observed distribution. Second, any differences in size or accessibility among cones would have affected their odds of being recorded (see *Cangiano et al. (2012)* for rod vs cone recording bias).

Third, compared to the classical approach of estimating cone spectral preference by adjusting the strengths of flashes of different wavelengths to match their response amplitudes, our use of the ratio of the peak response amplitude to moderately bright G and UV flashes of equal strength (our G/UV ratios) shifts values somewhat toward unity as cone saturation is approached (we chose this approach because it was quick and avoided exposing rods to very bright light). Fourth, there could be a contribution of other cones coupled to the recorded cone, which would shift the measured G/UV ratio in one or the other direction depending on their relative expression of S- and M-opsins. Finally, we cannot exclude that in some experiments, coupled rods may have retained a small response to the bright flash following the saturating pre-flash, again shifting the G/UV ratio. These last two factors could explain the shifts in G/UV ratio observed in some cones during spontaneous coupling increase and superfusion with MFA (*Figure 9—figure supplement 2*), as well as in the 'reversal' experiment of *Figure 8B*.

### Functional range of rod–cone coupling and its relationship to the anatomical substrate

GJs are often documented only anatomically, leaving open the question of their functional impact on neurons. Moreover, since they can be actively regulated, knowledge of their full coupling potential is essential. Our finding that the amplitude of rod signals in cones may attain a significant fraction of their source amplitude is a measure of the importance that rod–cone GJs can have in visual processing by the cone pathway. Is such a strong level of coupling compatible with what is known about rod–cone GJs?

Each cone in mouse forms junctional contacts with an average of 32 rods (*Tsukamoto et al., 2001*). To our knowledge, the only measurement in mammals of the transjunctional conductance was made in rod–cone pairs of the cone-dominated ground squirrel retina (legend of supplementary Figure 6 in *Li et al., 2010*). Assuming that their average value of 121 pS for coupled pairs (corresponding to ~20 open homotypic Cx36 channels; *Moreno et al., 2005*) was also representative of the mouse, multiplying it by a rod–cone convergence of 32 gives a summed junctional conductance of ~3900 pS. This would imply that a 4 mV hyperpolarization in rods relative to a maximally coupled cone would be sufficient to draw from it an overall junctional current of ~16 pA—a very large value as it is comparable to estimates of the circulating current in mouse cones in darkness (*Nikonov et al., 2006*). Therefore, our data are compatible with the limited available evidence. Moreover, our results bear direct relevance to other mammals, including primates: in the *area centralis* of the cat retina, *Smith et al. (1986)* estimated a convergence of ~48 rods on each cone, while in the peripheral retina of macaque, ~25 rods converge on each cone (*O'Brien et al., 2012*), a remarkably similar value to that in the mouse.

We did not observe cone signals entering rods through GJs, since rods did not respond appreciably to bright cone-stimulating flashes delivered during a rod-saturating background, or following rod-saturating pre-flashes. Two possible explanations for this are: (*i*) rod–cone divergence in mouse is small (~1; *Tsukamoto et al., 2001*); (*ii*) coupling might not be promoted when patching on rods.

### Rod signals fed into cones are accelerated

A previous study in macaque reported a twofold speed-up of rod signals as they flow into cones through GJs (*Hornstein et al., 2005*). We confirmed this in mouse, observing however a much less dramatic effect: in patched cones, the time to peak (TTP) of dim flash responses was 197 ms (SD 35, n = 24), significantly shorter (p<0.0001) than the 258 ms (SD 25, n = 23) of rods recorded in loose seal (rod and cone data were from both the kinetics protocol and the spectral protocol at 24°C) (note that the TTP of dim flash responses in patched rods was 288 ms (SD 51, n = 24), moderately but significantly longer (p=0.02) than that of rods recorded in loose seal; since rods recorded in loose seal did not display a kinetics rundown, their TTP values were assumed to be representative of the unperturbed state of rods in our preparation). The considerable discrepancy between the magnitude of the speedup in macaque and mouse could be explained if macaque rods underwent a rundown of kinetics during recordings (as it occurs in mouse), without this being noticed.

### Mechanisms of the spontaneous increase in coupling

As mentioned in the 'Results', a progressive increase in junctional coupling as a specific consequence of patch recording was observed in other studies targeting Cx36-expressing neurons, including heterologous expression systems (*Zoidl et al., 2002*; *Del Corsso et al., 2012*) and rat retinal amacrine AII cells (*Veruki et al., 2008*). Veruki et al., recording in whole cell mode, found that high resistance

(i.e., fine-tipped) electrodes prevent the coupling increase, suggesting a dialysis process. *Del Corsso et al. (2012)* surmised that an increase in intracellular $Ca^{2+}$ triggers the coupling increase during whole cell recordings in neuroblastoma cells in vitro. Moreover, they reported that the coupling increase did not occur when recording in perforated patch clamp mode with amphotericin-B, again implicating pipette-cell dialysis.

However, here we reach a different conclusion from that of the studies cited above. First, in cones, this phenomenon is not dependent on diffusion of $Ca^{2+}$ between the cell and the pipette, as the spontaneous increase in coupling was expressed in perforated patch recordings, even in combination with EGTA in the pipette solution. This, therefore, excludes the possibility of a $Ca^{2+}$ entry through hypothetical $Ca^{2+}$-permeable pores formed by amphotericin-B in the patch membrane (*Romero et al., 2009*; see also 'Results'). Second, our recordings in whole cell patch clamp with buffered pipette $Ca^{2+}$ levels, made in close proximity to the GJs, strongly suggest that changes in free $[Ca^{2+}]_i$ in cones are not responsible for the expression of this phenomenon. Moreover, since the progressive increase in coupling was observed both when preserving the intracellular environment of the cone (perforated patch recordings) and when intentionally dialyzing it (whole cell recordings), a general tentative conclusion may be advanced: diffusible messenger molecules in cones are not necessarily involved in the process. Thus, any diffusible messengers (including $Ca^{2+}$) could be located in the rods surrounding the recorded cone and act on the connexon hemichannels on the rod side.

Evidence for a primary trigger mechanism can be found in a previous study on macaque (*Hornstein et al., 2005*), which reported that Neurobiotin injected in cones at the end of perforated patch recordings diffused preferentially to rods located under the recording pipette, leading them to suggest that 'the electrode might alter the coupling efficiency of the rod–cone junctions by mechanical disturbance'. This observation implies that their cones also expressed a progressive increase in coupling, a process whose electrophysiological counterpart must have gone unnoticed. In this scenario, and taking into account the arguments given above, coupling would be promoted by stretch/deformation of the rods adjacent to the recorded cone. This could explain the large variability in the rate of increase and maximum level of coupling that we observed in our recordings (*Figure 4*), since the angle of entry and depth of the pipette in the tissue varies from experiment to experiment, likely changing its mechanical interaction with neighboring rods.

With regard to the downstream pathways, *Del Corsso et al. (2012)* found in their heterologous expression system a $Ca^{2+}$-mediated enhancement in the activity of calmodulin-dependent protein kinase II (CaMKII), which would lead to phosphorylation of Cx36 and increased channel conductance. Interestingly, CaMKII was recently found to participate in the physiological regulation of Cx36 in AII amacrines (*Kothmann et al., 2012*), raising the possibility that this kinase may play a role also in photoreceptors, on the rod side of rod–cone GJs (see above).

## Candidate routes for the in vivo recruitment of rod–cone coupling

At the start of the recordings, we found most cones modestly coupled to rods relative to their full coupling potential, although we observed strong initial coupling in a limited sample (*Figure 4*). A key question that needs to be addressed is when the retina exploits the strong coupling potential detected in this study. Candidate regulators of rod–cone GJs in mammals are (references in the 'Introduction'): (*i*) the circadian synthesis and release of the endogenous retinal neuromodulators, melatonin, dopamine, and adenosine, whereby coupling would be stronger at night when melatonin levels are high; most mouse strains, including the C57BL/6 used here, have deficits in melatonin synthesis (*Roseboom et al., 1998*) and are thus inadequate to investigate circadian processes; it is therefore possible that the low level of coupling that we estimated for our unperturbed cones may be related to this deficit and that full blown coupling, of the strength emerging during our recordings, is recruited in melatonin-competent mouse strains at night; (*ii*) acute light exposure, acting on the same neuromodulators and possibly also locally in the photoreceptors, which would inhibit coupling.

Unfortunately, unresolved discrepancies persist in the literature, a major one being the lack of effect of light and dopamine on rod–cone coupling studied with patch clamp in macaque (*Schneeweis and Schnapf, 1999*). Similar discrepancies (*Hartveit and Veruki, 2012*) exist for the GJs between AII amacrines, also containing Cx36 and thought of being modulated in a circadian and light-dependent fashion. While a common explanation for these conflicting pieces of evidence may eventually be found, our demonstration that rod–cone coupling can be upregulated so as to have a major impact on

cones provides an important framework in which to place circadian/light-dependent modulation of these GJs in mouse. In fact, (*i*) let's suppose that the level attained in our experiments by the fully developed coupling process had been quite smaller than what we actually observed; then this would have raised serious doubts about the physiological importance of circadian/light-dependent modulation of rod–cone coupling for mouse visual processing; (*ii*) given however our demonstrated high level of coupling, should a future direct examination of the role of circadian rhythmicity find that only a small fraction of this coupling potential is utilized, one might then well suspect that yet unidentified physiological factors play a greater role in recruiting rod–cone GJs, or, alternatively, that these may have an important part in response to stress and injury. One should recollect that GJs play a dual role, for example, conduits of electrical signals and of intracellular molecules. Cx36, in particular, is critically involved in neuronal responses to injury and disease (*Belousov and Fontes, 2013*). Currently, only a few studies have investigated this in the retina, with both positive (*Striedinger et al., 2005*; *Paschon et al., 2012*) and negative results (*Kranz et al., 2013*) depending on the specific models tested.

The existence of a large degree of GJ-mediated anatomical convergence from rods to cones is not peculiar of the mouse, since it has been found in other rod-dominated mammalian retinas including the peripheral retina of macaque. It follows that our findings on the functional impact of rod input in mouse cones may have broad relevance, supporting the possibility that rod–cone coupling plays a significant role in vision and/or in biochemical signaling between photoreceptors.

## Materials and methods

Except when stated otherwise, dissection and recordings were performed as previously described (*Cangiano et al., 2007*, *2012*). Briefly, adult C57BL/6J mice, raised on a 12/12 hr light cycle, were dark-adapted for 3–5 hr and recorded during the subjective late afternoon of the animal. Dissection and slicing were done in chilled bicarbonate-buffered Ames' medium (A1420; Sigma-Aldrich, St. Louis, MO, USA) under dim far red illumination. Blind patch recordings of photoreceptors (at 24°C or 36°C) were also done in Ames' medium and targeted sections of retinal slices (250 µm) corresponding to both the central and peripheral retina. We could not identify the position of our recordings along the dorsoventral axis of the retina. Perforated patch clamp and whole cell recordings were obtained from both rods and cones, while loose seal recordings could be obtained only from rods. Several intracellular solutions were used in this study, all including 0.5 mg·ml$^{-1}$ Lucifer Yellow and corrected to a pH of 7.20 with KOH/HCl. For perforated patch, the backfilling solution also contained 0.4 mg·ml$^{-1}$ Amphotericin-B pre-dissolved in DMSO at 60 mg·ml$^{-1}$. *Standard perforated patch*: (in mM) 90 Kaspartate, 20 $K_2SO_4$, 15 KCl, 10 NaCl, 5 $K_2$Pipes. *EGTA zero $Ca^{2+}$ perforated patch*: 80 Kaspartate, 20 $K_2SO_4$, 25 KCl, 10 NaCl, 5 $K_2$Pipes, 1 EGTA, 1 $MgCl_2$. *EGTA low $Ca^{2+}$ whole cell*: 90 Kaspartate, 20 $K_2SO_4$, 10 KCl, 5 NaCl, 5 $K_2$Pipes, 4 $Na_2$ATP, 0.1 NaGTP, 5 EGTA, 4.2 $MgCl_2$, 1.1 $CaCl_2$. *BAPTA low $Ca^{2+}$ whole cell*: 74.5 Kaspartate, 20 $K_2SO_4$, 10 KCl, 5 NaCl, 5 $K_2$Pipes, 4 $Na_2$ATP, 0.1 NaGTP, 7 $K_4$BAPTA, 4.2 $MgCl_2$, 1.15 $CaCl_2$. *EGTA high $Ca^{2+}$ whole cell*: 90 Kaspartate, 20 $K_2SO_4$, 10 KCl, 5 NaCl, 5 $K_2$Pipes, 4 $Na_2$ATP, 0.1 NaGTP, 5 EGTA, 4.2 $MgCl_2$, 4.8 $CaCl_2$. The low and high calcium whole cell solutions were estimated to provide free $Ca^{2+}$ concentrations of 50 nM and (approximately) 5 µM, respectively, and a free $Mg^{2+}$ concentration of ~0.7 mM (Maxchelator software; Ca-Mg-ATP-EGTA Calculator v1.0 and WebMaxC standard). Based on an analysis of the liquid junction and Donnan potentials in our perforated-patch recordings (*Cangiano et al., 2012*), we report uncorrected values of the membrane potential. In some experiments, GJs were blocked with meclofenamic acid (Sigma-Aldrich).

### Light stimulation

Full field stimuli of unpolarized light were delivered by a green LED (peak emission at 520 nm; OD520; Optodiode Corp., Newbury Park CA) or an ultraviolet LED (peak emission at 365 nm; APG2C1-365-S; Roithner LaserTechnik, Vienna Austria) mounted beside the objective turret. LEDs were driven by current sources commanded through the analog outputs of a Digidata 1320A (Axon Instruments, Foster City, CA). The power density reaching the recording chamber vs LED drive was measured separately with a calibrated low power detector (1815-C/818-UV; Newport, Irvine, CA) positioned at the recording chamber. Flash duration was in the range of 1–10 ms. Unless specified in the protocol, consecutive bright flashes were delivered at intervals of 12 s or more between each other. The photon flux density reaching the photoreceptors was derived from the measured power density and was likely to be overestimated to varying degrees across recorded cells due to reflection

at the air–water interface and absorption by the surrounding tissue, including retinal pigment epithelium. Outer segments could be oriented at a variety of angles with respect to the direction of incident light.

Rod input in cones was dissected with a *kinetics protocol* and a *spectral protocol*. These consisted of a mix of dim and bright flashes at 520 nm (green, G) or 365 nm (ultraviolet, UV). For reference, in mouse, the absorption peaks of rhodopsin, M-opsin, and S-opsin are at 498 nm, 508 nm, and 359 nm, respectively (*Sun et al., 1997*; *Yokoyama et al., 1998*). Dim flashes (16.6 photons·µm$^{-2}$) were sufficient to elicit a significant response in rods (*Figure 1A,B*, curves on the left; our data, see legend) but expected to be too weak to directly stimulate cones (*Figure 1A,B*, curves on the right; predictions from the literature, see next paragraph). Bright flashes (bright$_a$ = 1570, bright$_b$ = 3140 ph·µm$^{-2}$) were sufficiently strong to saturate rods for 1–2 s (*Figure 1A*, lower graph, our data) and expected to evoke a measurable response in cones (UV flashes in all cones, while G flashes in most cones; *Figure 1A,B*, curves on the right; predictions from the literature, see below).

Most mouse cones express both S and M opsins, with their relative proportions varying along the vertical axis of the retina (M dominates dorsally, while S dominates in the mid and ventral retina) (*Applebury et al., 2000*; *Haverkamp et al., 2005*; *Nikonov et al., 2006*; *Daniele et al., 2011*; *Wang et al., 2011*). In *Figure 1A,B*, we show the predicted average flash response profiles of cones at different retinal latitudes (1–4: mixed S/M cones) and that of the small but distinct population of pure S cones sparsely distributed throughout the retina. To generate these profiles, we: (1) used hyperbolic saturation functions (*Nikonov et al., 2006*); (2) assumed the flash sensitivity of the normalized response of S- and M-cones in G$_t\alpha^{-/-}$ mice, recorded with suction pipette and without a rod-saturating background (Table 1 in *Nikonov et al., 2006*: 0.042% photons$^{-1}$·µm$^2$, equivalent to a half-saturating flash strength of 2381 ph·µm$^{-2}$), to be valid for S- and M-cones in wild type mice; moreover, we assumed that all cones express the same total amount of opsin; (3) the typical fraction of M-opsin expressed by cones as a function of their retinal latitude (i.e., the average among cones within each horizontal slice of retina) was taken as the average of the values predicted by two recently published quantitative models (*Daniele et al., 2011*; *Wang et al., 2011*) obtained from mice in which rod responses were absent or suppressed; since these models extend to slightly different retinal latitudes (±2.5 mm and ±2.0 mm, respectively), we normalized their ranges prior to averaging; the relevant parameters of the cones shown in *Figure 1*, given as %M-opsin/position, are **1**: 64%/dorsalmost, **2**: 24%/dorsal third, **3**: 4.7%/ventral third, **4**: 1.2%/ventralmost, **S**: 0%/ubiquitous; (4) the absorbance of M-opsin at 365 nm (peak of the β-band) was taken as 20% of maximum, while for S opsin, its absorbance at 520 nm was taken to be 4 log-units below maximum (based on *Govardovskii et al., 2000*).

The kinetics protocol (*Figure 1C*) was designed to detect rod input in cones by exploiting the high sensitivity of rods and their slow recovery after a saturating flash. A dim G test flash was delivered both before a bright$_a$ G flash and at three increasing delays following it (1, 2.5, 4 s). Observing in cones: (*i*) a response to the first dim flash, (*ii*) a slow plateau after the bright flash, and (*iii*) a progressive recovery of the dim flash response, would be evidence for rod coupling. The spectral protocol (*Figure 1D*) was designed to determine the impact of rod coupling on the spectral preference of cones and the possible relationship between cone opsin expression and coupling strength. The cones' apparent spectral preference was determined by delivering G and UV flashes of the same strength (equal number of ph·µm$^{-2}$; either dim or bright$_b$ flashes). The cones' intrinsic preference was determined by delivering G and UV bright$_b$ flashes after a G rod-saturating pre-flash.

## Data analysis

Data are reported as mean and SD (standard deviation), SEM (standard error of the mean). Statistical significance was assessed with the Mann–Whitney–Wilcoxon test. In all figures, electrophysiological records were 'box car' filtered with a running window of 20 ms.

## Acknowledgements

We thank Prof Luigi Cervetto for his continued support throughout this project and for providing critical comments to the manuscript. We also thank Prof Alberto Cangiano for providing helpful comments on the manuscript.

# Additional information

### Funding

| Funder | Author |
| --- | --- |
| University of Pisa | Lorenzo Cangiano |

The funder had no role in study design, data collection and interpretation, or the decision to submit the work for publication.

### Author contributions

SA, Conception and design, Acquisition of data, Analysis and interpretation of data, Drafting or revising the article; CG, Critically revised the article, Contributed unpublished essential data or reagents; LC, Conception and design, Analysis and interpretation of data, Drafting or revising the article, Contributed unpublished essential data or reagents

### Ethics

Animal experimentation: All procedures involving the handling of experimental animals were approved by the Ethical Committee of the University of Pisa (prot. n. 2891/12) and were conducted in accordance with Italian (D.lgs.vo 116/92) and EU regulations (Council Directive 86/609/EEC).

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
