## [Decision Letter]

Thank you for sending your work entitled “Mouse cones have a striking coupling potential to rods” for consideration at *eLife*. Your article has been favorably evaluated by a Senior editor and 3 reviewers, one of whom is a member of our Board of Reviewing Editors.

This is a very nice paper describing gap junctional coupling between mammalian rods and cones. A good deal of indirect evidence, and a few direct experiments, supports a picture in which rod signals can traverse the retina through a primary (rod bipolar) pathway and a secondary (rod–cone gap coupling) pathway. The present paper provides to date the most detailed and complete characterization of the cone signals inherited from rods via gap junctions. Building on work presented in the authors' previous paper presenting the first whole-cell recordings from mouse cones (6), it makes a compelling case that the gap-junctional pathway for rod signals through cones exists in mouse, and characterizes a number of aspects of the pathway.

The paper presents patch-clamp recordings with the perforated-patch technique from cones in slices of mouse retina. The experiments seem ably done. The authors make the interesting observation that rod input into cones can be modest at the beginning of the recording but can become large as the recording proceeds, presumably as the result of some change in the physiology of the photoreceptors or retina. Although the paper leaves open the reason for the large change in gap-junctional conductance, the simple demonstration of large rod input into cones – regardless of the conditions of the recording – provides compelling evidence that junctions between rods and cones in mouse are capable of transferring large voltage signals.

Overall, the work is carefully executed with statistical sampling and analyses that support the primary conclusions. In general the conclusions drawn are well supported. This paper is likely to interest many investigators in photoreceptor and retinal research. The reviewers think that it will be a strong addition to *eLife*.

A few specific areas that could use attention follow.

1) One major problem arises from the fact that the mouse cone S- and M-opsins are expressed in dorso-ventral counter gradients, and co-expressed in most cones, and the authors have not made their recording from well identified regions of the retina. From the limitations of their dissection methodology, it follows that any general inferences to be drawn about the relative sensitivity of cones to excitation of S- and M-opsins are not very informative, and in some cases misleading. The authors do state that they took tissue samples “randomly” from different regions of the retina, but it seems that this really mean that they just didn't pay any attention to the region of the retina from which their tissue samples were taken. But even if they could argue that the tissue samples were truly randomly sampled from all retinal areas, they would need to provide a much clearer analysis (e.g., based on the S/M opsin gradients in [2]; Yang et al., 2011; [8]) of the predictions of the S/M sensitivity of cones for their sampling method.

Minimally, what the authors must do is be completely up-front about the problems of the dorso-ventral gradients of cone opsin expression, and address how this might have affected their results and conclusions. This problem does not impact the major conclusions of the paper about rod signaling through cones, but rather mainly leads to issues and discussion irrelevant to the main point of the paper.

The authors should consider deleting the last three sections of Results. The “Mixed S/M” section is inappropriate to the Results; it might be incorporated into the Discussion if the authors could shorten it significantly. As written, however, this section is diffuse and leads to no clear conclusion. The section “Rod signals” is again inappropriate to the Results section but could perhaps be incorporated into the Discussion.

2) A second major issue concerns the modeling of the “resistive” network of the cones and rods (Figure 8; Methods). The model is superficial, in part because too many of the conductances required for generating a proper model are unknown, and in part because it neglects a number of important and well established electrophysiological features of rods and cones, including hyperpolarization-activated conductance (Ih currents, HCN channels) and the voltage dependence of the cone cGMP-activated conductance (treated as not voltage-dependent). This section in the Results (including the model and Figure 8) should simply be deleted. It is premature for the authors to make a model of rod/cone conductance when there remain so many uncertainties, including the reason for the large variability from one cone to the next and the change in the properties of the junctional conductance with time of recording. The model should be introduced in a future publication, once the nature of rod/cone coupling in mouse is better understood and more clearly quantified.

3) The cause of the gradual shift of cones' signaling rod-like responses is not resolved in the paper. In the discussion the authors suggest it arises from increased [Ca^2+^]_i_ arising “as a consequence of stretch/deformation of the cell membrane in the pipette”. The authors should attempt an experiment in which they intentionally elevate [Ca^2+^]_i_ via the patch pipette, and see if this accelerates the presumptive GJ conductance increase.

---

## [Author Response]

*1) One major problem arises from the fact that the mouse cone S- and M-opsins are expressed in dorso-ventral counter gradients, and co-expressed in most cones, and the authors have not made their recording from well identified regions of the retina. From the limitations of their dissection methodology, it follows that any general inferences to be drawn about the relative sensitivity of cones to excitation of S- and M-opsins are not very informative, and in some cases misleading. The authors do state that they took tissue samples “randomly” from different regions of the retina, but it seems that this really mean that they just didn't pay any attention to the region of the retina from which their tissue samples were taken. But even if they could argue that the tissue samples were truly randomly sampled from all retinal areas, they would need to provide a much clearer analysis (e.g., based on the S/M opsin gradients in*
[2]*; Yang et al., 2011;*
[8]*) of the predictions of the S/M sensitivity of cones for their sampling method*.

*Minimally, what the authors must do is be completely up-front about the problems of the dorso-ventral gradients of cone opsin expression, and address how this might have affected their results and conclusions. This problem does not impact the major conclusions of the paper about rod signaling through cones, but rather mainly leads to issues and discussion irrelevant to the main point of the paper*.

We have introduced a new section in the Discussion titled “Cone opsin expression gradients and rod-cone coupling”, which explicitly discusses the limitations of our dissection and recording techniques in the context of the S- and M-opsin expression gradients. We have also revised Figure 1 to account for the quantitative models of opsin expression gradients in [8] and [56].

*The authors should consider deleting the last three sections of Results. The “Mixed S/M” section is inappropriate to the Results; it might be incorporated into the Discussion if the authors could shorten it significantly. As written, however, this section is diffuse and leads to no clear conclusion*.

We have removed these sections from the Results, incorporating a limited portion of this text in a new section of the Discussion titled “Cone opsin expression gradients and rod-cone coupling”.

*The section “Rod signals” is again inappropriate to the Results section but could perhaps be incorporated into the Discussion*.

We have followed the reviewers’ suggestion by removing this section from the Results and incorporating it, with some minor editing, in the Discussion.

*2) A second major issue concerns the modeling of the “resistive” network of the cones and rods (*Figure 8*; Methods). The model is superficial, in part because too many of the conductances required for generating a proper model are unknown, and in part because it neglects a number of important and well established electrophysiological features of rods and cones, including hyperpolarization-activated conductance (Ih currents, HCN channels) and the voltage dependence of the cone cGMP-activated conductance (treated as not voltage-dependent). This section in the Results (including the model and*
Figure 8*) should simply be deleted. It is premature for the authors to make a model of rod/cone conductance when there remain so many uncertainties, including the reason for the large variability from one cone to the next and the change in the properties of the junctional conductance with time of recording. The model should be introduced in a future publication, once the nature of rod/cone coupling in mouse is better understood and more clearly quantified*.

We have removed from the Results the section titled “The summed junctional conductances to rods can exceed the cone’s light-sensitive conductance”, as well as Figure 8 and the corresponding section in the Materials and methods.

*3) The cause of the gradual shift of cones' signaling rod-like responses is not resolved in the paper. In the discussion the authors suggest it arises from increased [Ca*^*2+*^*]*_*i*_
*arising “as a consequence of stretch/deformation of the cell membrane in the pipette”. The authors should attempt an experiment in which they intentionally elevate [Ca*^*2+*^*]*_*i*_
*via the patch pipette, and see if this accelerates the presumptive GJ conductance increase*.

By performing a new set of experiments, we have addressed the hypothesis that increased calcium in cones is responsible for the progressive increase in coupling observed during patch recordings. We have approached this question by obtaining whole cell patch clamp recordings from cone synaptic terminals, to dialyze cones in the immediate proximity of their GJs to rods, using low (∼50 nM free [Ca^2+^]_i_; n=6) or high calcium buffered intracellular solutions (approx. 5 µM free [Ca^2+^]_i_; n=6). Our findings are presented in a new section of the Results and in a new Figure 11. In summary, we have found that the process of coupling increase is expressed even when free [Ca^2+^]_i_ in the pedicle is clamped to low levels, both with EGTA and BAPTA. Conversely, high calcium levels did not appear to favor coupling compared to the low calcium condition or to perforated patch recordings.

Our conclusions, stated in a revised Abstract and Discussion, are that calcium changes on the cone side of the GJs appear not to be responsible for the progressive increase in coupling. Moreover, since this phenomenon was expressed both in perforated patch and whole cell recordings, two techniques at opposite extremes in terms of their impact on the intracellular milieu of the cone, we suggest that any diffusible messenger involved in the process would likely be located in the neighboring rods.